# Assessing FY-3E HIRAS-II Radiance Accuracy Using AHI and MERSI-LL

## Hongtao Chen and Li Guan *

Collaborative Innovation Center on Forecast and Evaluation of Meteorological Disasters, Nanjing University of Information Science &Technology, Nanjing 210044, China
* Correspondence: liguan@nuist.edu.cn

**Abstract:** The FY-3E/HIRAS-II (Hyperspectral Infrared Atmospheric Sounder-II), as an infrared hyperspectral instrument onboard the world's first early morning polar-orbiting satellite, plays a major role in improving the accuracy and timeliness of global numerical weather predictions. In order to assess its observation quality, the geometrically, temporally, and spatially matched scene homogeneous HIRAS-II hyperspectral observations were convolved to the channels corresponding to the Himawari-8/AHI (Advanced Himawari Imager) and FY-3E/MERSI-LL (Medium-Resolution Spectral Imager) imagers from 15 March to 21 April 2022, and their brightness temperature deviation characteristics were statistically calculated in this paper. The results show that the HIRAS-II in-orbit observed brightness temperatures are slightly warmer than the AHI observations in all the matched AHI channels (long wave infrared channel 8 to channel 16) with a mean brightness temperature bias less than 0.65 K. The bias of the atmospheric absorption channel is slightly larger than that of the window channel. A standard deviation less than 0.31 K and a correlation coefficient higher than 0.98 in all channels means that the quality of the observation is satisfactory. The thresholds chosen for the colocation approximation factors (e.g., observation geometry angle, scene uniformity, observation azimuth, and observation time) for matching the HIRAS-II with AHI contribute little and negligible uncertainty to the bias assessment, so the difference between the two observed radiations is considered to be mainly from the systematic bias of the two-instrument measurement. Compared with MERSI-LL window channel 5, the observations of both instruments are very close, with a mean bias of 0.002 K and a standard deviation of 0.31 K. The mean brightness temperature bias (HIRAS-II minus MERSI-LL) of the MERSI-LL water vapor channel 4 is 0.66 K with a standard deviation of 0.22 K. The mean brightness temperature bias of channel 6 and channel 7 is 0.63 K (the standard deviation is 0.36 K) and 0.5 K (the standard deviation is 0.3 K), respectively. The biases of channel 4 are significantly and positively correlated with the target scene temperature, and the biases of channel 6 and 7 show a U-shaped change with the increase in the scene temperature, and the biases are smallest (close to 0 K) when the scene temperature is between 250 K and 280 K. The statistical characteristics of the HIRAS-II–MERSI-LL difference vary minimally and almost constantly over a period of time, indicating that the performance of the HIRAS-II instrument is stable.

**Keywords:** Hyperspectral Infrared Atmospheric Sounder-II; FY-3E; Himawari-8/AHI

## 1. Introduction

Satellite-borne infrared hyperspectral atmospheric sounders can obtain global meteorological observations with high precision and high spectral resolution and have been frequently applied to retrieving atmospheric temperature and humidity profiles, data assimilation, and climate studies [1]. In addition, quality control of meteorological satellite observations is a pivotal step before using satellite data for assimilation and retrieval, and it is the fundamental basis for building long-term infrared hyperspectral datasets [2].

FY-3E, the world's first early morning polar-orbiting meteorological satellite, was successfully launched on 5 July 2021. It effectively fills a void for global satellite observations

and provides 100% global satellite data coverage for numerical weather prediction (NWP) at 6-h intervals [3]. The HIRAS-II (Hyperspectral Infrared Atmospheric Sounder-II) is a continuation of the infrared hyperspectral instrument HIRAS onboard the FY-3D, and it can provide hyperspectral observations in the thermal infrared band with 3041 contiguous channels. Compared with its predecessor, the field of view (FOV) array within a field of regard (FOR) has changed from 2 × 2 to 3 × 3 with the spatial resolution increased from 16 km to 14 km at the nadir, the sensitivity is enhanced by more than 2 times with the spectral calibration accuracy increased by 30% and the radiometric calibration accuracy increased from 0.7 K to 0.5 K [4]. HIRAS-II is expected to become the reference instrument for infrared remote sensing instruments; therefore, independently assessing its data quality for radiance measurements is of great importance in improving the accuracy and timeliness of global numerical weather prediction.

The combined remote sensing and intercalibration based on the satellite-borne infrared hyperspectral atmospheric sounder and high-spatial-resolution imager has become one of the most effective means to quantify the radiometric calibration accuracy for both types of instruments. Gunshor calibrated the water vapor channels and window channels of five geostationary satellite imagers using the High-Resolution Infrared Radiation Sounder (HIRS) and the Advanced Very-High-Resolution Radiometer (AVHRR) onboard NOAA-14 [5]. Tobin used the Atmospheric Infrared Sounder (AIRS) to evaluate the radiometric accuracy of the Moderate-Resolution Imaging Spectroradiometer (MODIS) carried on the same platform [6]. Wang used the Infrared Atmospheric Sounding Interferometer (IASI) to intercalibrate the water vapor channel of the GOES-11 and GOES-12 [7]. Xu Na et al., using the hyperspectral measurements of IASI as a reference, objectively assessed the on-orbit radiometric calibration accuracy of the FY-3A Medium-Resolution Spectral Imager (MERSI) thermal infrared channel [8]. Gong used the Cross-track Infrared Sounder (CrIS) onboard the Suomi National Polar-orbiting Partnership (SNPP) satellite platform to cross-check the thermal infrared channels of the Visible Infrared Imaging Radiometer Suite (VIIRS) on the same platform [9]. Yang et al. assessed the relative bias of the HIRAS radiometric calibrations using the Metop-A/B IASI observations based on the Simultaneous Nadir Overpass (SNO) intercalibration method [1].

Accuracy assessments of satellite instrument on-orbit calibrations is necessary to ensure product consistency and interoperability, and it is also extremely important for bias correction in data assimilations [10]. However, HIRAS-II—which is on board the first early morning polar-orbiting satellite launched last year—is operational this year, and the quality of its radiance measurements has not yet been reported in the literature. The Advanced Himawari Imager (AHI) mounted on the Japanese geostationary meteorological satellite Himawari-8 is recognized as one of the most accurate imaging instruments in the world. The AHI is greatly improved over those of the MTSAT (Multi-functional Transport Satellite) series in terms of the number of bands, spatial resolution, and temporal frequency; and infrared (IR) band calibration is accurate to within 0.2 K with no significant diurnal variation [11–13]. Therefore, this paper evaluates the quality of the radiance measurements based on the spatially and temporally matched Himawari-8/AHI observations from 15 March to 21 April 2022, and also performs an intercomparison with the thermal infrared observations of MERSI-LL carried out on the same platform.

## 2. Data Used in the Research

The HIRAS-II is an interferometric Fourier transform spectrometer carried in a polar orbit 836 km above the ground. HIRAS-II views the ground in the conventional mode through a cross-track rotary scan mirror that provides ±50.4° ground coverage every 8 s. Each scan line observes 32 fields of regard (FORs), including 28 continuous Earth targets, 2 cold space targets, and 2 blackbody targets on the satellite. Each field of regard (FOR) includes a 3 × 3 field of view (FOV) with a spatial resolution of 14 km at the nadir. HIRAS-II covers the 3.92–15.38 µm infrared band with 3041 continuous channels at a spectral resolution of 0.625 cm$^{-1}$. The HIRAS-II Level 1 radiance observations from 15 March to 21

April 2022 are used in this paper. The data can be found on the Chinese Feng Yun satellite remote sensing data service network (http://data.nsmc.org.cn accessed on 11 April 2022).

The Moderate-Resolution Spectral Imager-Low Light (MERSI-LL) is an important optical instrument onboard FY-3E with microlight and infrared detection capabilities. It is equipped with one visible channel operable with low-level illumination and six infrared channels. The spatial resolution of the two infrared split-window channels is 250 m, and the remaining channels are 1000 m. The MERSI-LL Level 1 radiance observations with a spatial resolution of 1000 m from 15 to 22 March 2022 are used in this paper and can be downloaded from http://data.nsmc.org.cn (accessed on 11 April 2022).

The AHI on the geostationary satellite Himawari-8 successfully launched in October 2014 and is one of the most advanced spaceborne imagers in the world. It has 16 observation channels (3 visible, 3 near-infrared, and 10 infrared), of which the spatial resolution of the infrared channel is 2 km and the temporal resolution is 10 min. Himawari-8/AHI radiation data obtained from the Japan Earth Observation Data Center (https://www.eorc.jaxa.jp/ptree/index.html accessed on 11 April 2022) from 15 March to 21 April 2022 are analyzed.

The channel settings and performance of the thermal infrared band covered by HIRAS-II, AHI, and MESI-LL are shown in Table 1. The last row of the table (spectral coverage) specifies the central wavelength and the corresponding peak height of the weighting function (in parentheses) for each channel. The AHI has nine channels that can be completely spectrum matched with the HIRAS-II spectrum, of which channels 8, 9, and 10 are water vapor absorption channels; channels 13, 14, and 15 are window channels; and channels 11, 12, and 16 are $SO_2$, $O_3$, and $CO_2$ absorption channels, respectively. The weighting function peak heights of AHI channels 8, 9, 10, 12, and 16 are 300 hPa, 371 hPa, 532 hPa, 40 hPa, and 863 hPa, respectively. The weighting function heights of the remaining channels are almost near the surface. For MERSI-LL, only channels 4, 5, 6, and 7 can be completely spectrum matched. Channel 4 is a water vapor channel (the peak height of the weighting function is approximately 400 hPa), and channels 5, 6, and 7 are window channel with a central wavelength of 8.55 µm, 10.8 µm, and 12.0 µm, respectively.

**Table 1.** Instrument performance parameters of HIRAS-II, AHI, and MERSI-LL in the longwave infrared band.

|  | **HIRAS-II** | **AHI** | **MERSI-LL** |
|---|---|---|---|
| Satellite platform | FY-3E | Himawari-8 | FY-3E |
| Spatial resolution/km | 14 km (at nadir) | Infrared: 2 km | Infrared: 1 km |
| Spectral coverage | 3041 channels (3.92–15.38 µm) with spectral resolution 0.625 cm$^{-1}$ | Ch8: 6.2 µm (300 hPa) Ch9: 6.9 µm (371 hPa) Ch10: 7.3 µm (532 hPa) Ch11: 8.6 µm (window) Ch12: 9.6 µm (40 hPa) Ch13: 10.4 µm (window) Ch14: 11.2 µm (window) Ch15: 12.4 µm (window) Ch16: 13.3 µm (863 hPa) | Ch4: 7.2 µm (400 hPa) Ch5: 8.55 µm (window) Ch6: 10.8 µm (window) Ch7:12.0 µm (window) |

## 3. Observation Data Matching and Evaluation Method

Four major steps are involved in an intercomparison of the FY-3E/HIRAS-II infrared hyperspectral observations with the Himawari-8/AHI radiance observations of the corresponding longwave infrared channel, including (1) spectral convolution, (2) observation geometry and temporal matching, (3) spatial matching, and (4) uniformity checking. The matching process is shown in Figure 1. The following will be a step-by-step detailed introduction.

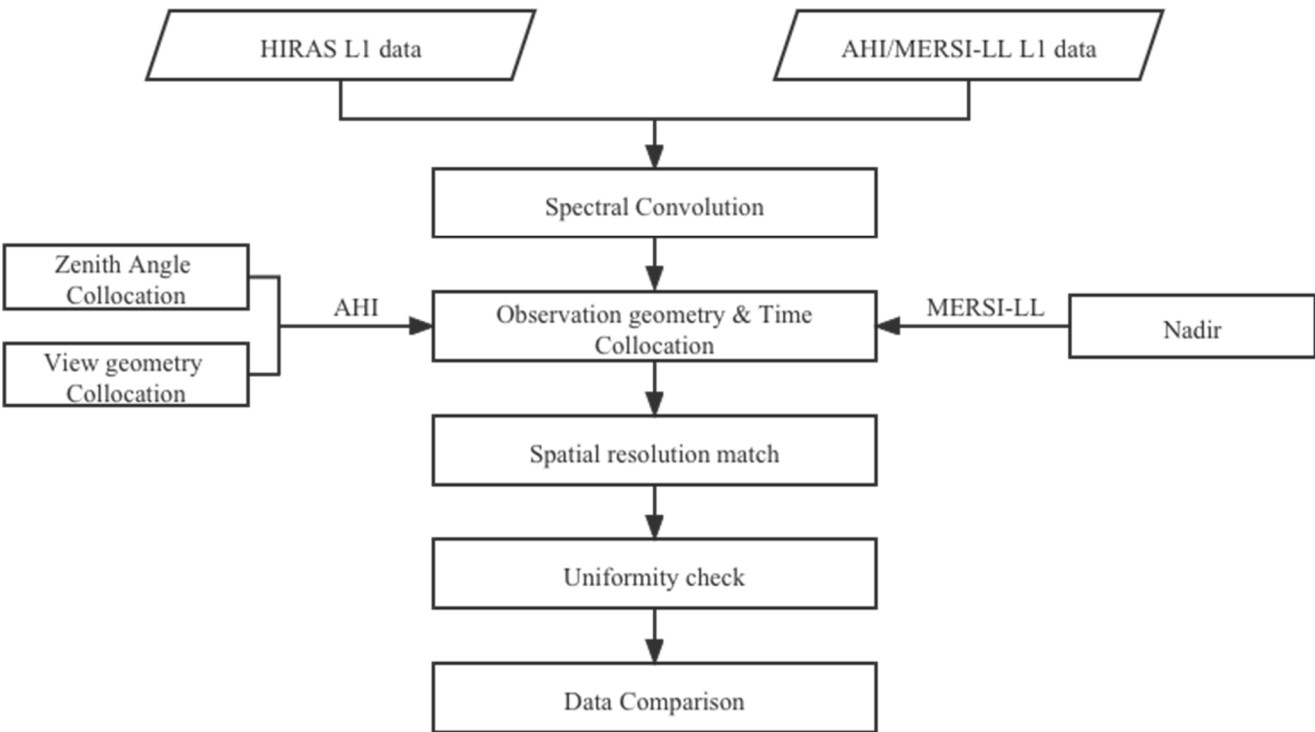

**Figure 1.** Intercomparison flow chart of HIRAS-II, AHI, and MERSI-LL.

### 3.1. Spectral Convolution

The most fundamental problem in the intercomparison between the spaceborne imager and sounder observation is radiance spectral matching. The two types of instruments with different spectral resolutions cannot be directly compared. To be compared with the imager instrument observations, the hyperspectral radiance must be convolved to match the spectral response function (SRF) of the broadband imager [14,15]. The computational formula to achieve spectral convolution is

$$L = \frac{\int_{\nu_1}^{\nu_2} R(\nu)S(\nu)d\nu}{\int_{\nu_1}^{\nu_2} S(\nu)d\nu} \tag{1}$$

where $\nu$ is the wavenumber, L is the radiance calculated by convolution, $R(\nu)$ is the hyperspectral radiance at the corresponding wavenumber and $S(\nu)$ is the spectral response function of the imager. The simulated HIRAS-II brightness temperature spectrum using fast radiative transfer mode RTTOV under American standard atmospheric conditions, as well as the AHI and MERSI-LL spectral response functions (SRF), are given in Figure 2. The bottom horizontal coordinate is the wavelength, the upper horizontal coordinate is the wavenumber, the left vertical coordinate is the simulated brightness temperature of HIRAS-II, and the right vertical coordinate is the spectral response function. The solid line in the figure is the spectral response function of the matched AHI channels 8–16, and the dashed line denotes the spectral response function of MERSI channels 4–7.

### 3.2. Observation Geometry and Temporal Matching

The most important step in the intercomparison between the imager and the sounder observation is to find the consistent (same) FOV [16]. The polar-orbiting satellite FY-3E passes the geostationary satellite Himawari's nadir 140.7 E at approximately 0830 or 2030 UTC daily, the observation time difference remained within 10 min. Since the measurements of the cross-track scanning instruments are sensitive to the scan angle (satellite zenith angle), the scan angle of both instruments in the matched field of view is specified to be less than 5° to ensure that HIRAS-II and AHI observe the same scene. To

minimize the difference in observation geometry and ensure that both instruments have similar observation geometry paths, the scan angle is further constrained [7,17]

$$\left| \frac{\cos\theta_1}{\cos\theta_2} - 1 \right| < 0.002 \tag{2}$$

where $\theta_1$ and $\theta_2$ are the scan angles of the geostationary and polar-orbiting satellites, respectively.

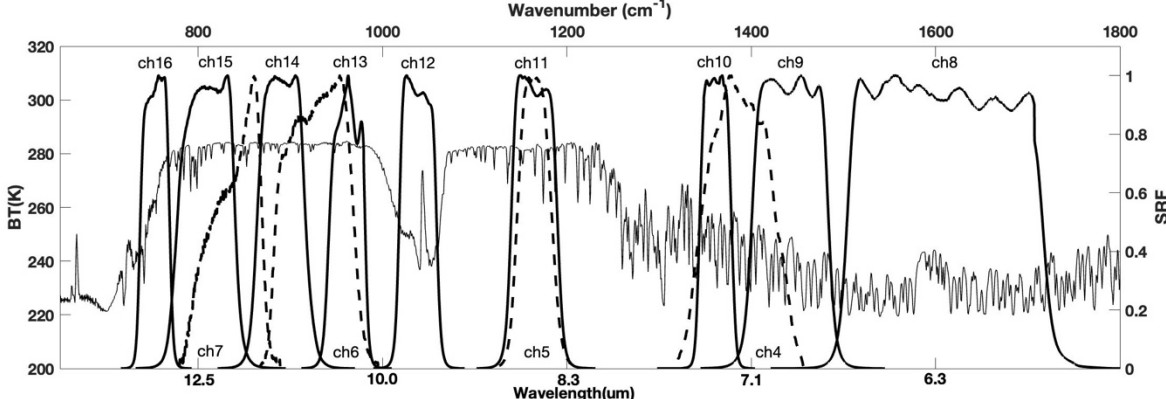

**Figure 2.** Simulated HIRAS-II brightness temperature spectra from RTTOV overlaid the spectral response functions of AHI (solid line) and MERSI-LL (dashed line), channel numbers for AHI are shown near the top of the graph and channel numbers for MERSI−LL are near the bottom.

### 3.3. Spatial Matching and Uniformity Check

The spatial field of view collocation of the FY-3E/HIRAS-II and Himawari-8/AHI in the infrared channel is shown in Figure 3, where the grid is the AHI pixel, the large circle is a HIRAS-II field of regard (FOR) centered on the field of view (FOV) to be matched (target) and the small circle is the HIRAS-II FOV within its FOR. The spatial resolution of the HIRAS-II FOV (14 km at the nadir) is coarser than that of the AHI infrared channel (2 km). Approximately 7 × 7 AHI pixels (small squares in Figure 3) fall in a HIRAS-II FOV (the small circle in Figure 3). The average of all these AHI pixel measurements is taken as the imager's measurement in the matching FOV [18]. This requires that the observation target be relatively homogeneous. Since there are many AHI pixels collocated in the HIRAS-II FOV, some of which are clear sky and some are cloudy, at the same time the underlying surface of the field of view is not homogeneous, it is necessary to check the uniformity of each HIRAS-II FOV, which is carried out by using the ratio of the standard deviation of the matched AHI radiations to its mean value

$$\text{Std}_{\text{fov}}/\text{Mean}_{\text{fov}} < 0.01 \tag{3}$$

where $\text{Std}_{\text{fov}}$ denotes the standard deviation of the observed radiance from all the matched AHI pixels within each HIRAS-II FOV, and $\text{Mean}_{\text{fov}}$ is the mean radiance of all these AHI pixels. The threshold value is set to 0.01. Only uniform scenes are selected for intercomparison to reduce the uncertainty introduced by the field of view averaging.

Furthermore, a constraint of background environmental uniformity of the HIRAS-II FOV is needed to compensate for the minor error of the spatial collocation, as well as to reduce the uncertainty due to different azimuths. The 3 × 3 HIRAS-II FOV (the large circle in Figure 3) centered on the HIRAS-II FOV to be matched is considered the background environment area.

$$\text{Std}_{\text{env}}/\text{Mean}_{\text{env}} < 0.05 \tag{4}$$

where $\text{Std}_{\text{env}}$ denotes the standard deviation of all the matched AHI pixels' radiance within the background area, and $\text{Mean}_{\text{env}}$ denotes the mean radiance of these matched AHI pixels.

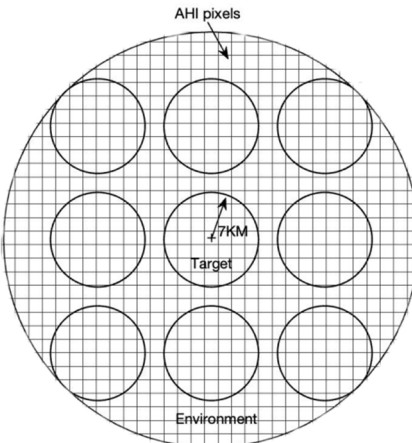

**Figure 3.** Spatial matching of HIRAS-II and AHI. The small square represents the AHI pixel, the small circle is the HIRAS-II FOV, and the large circle is the HIRAS-II FOR.

*3.4. Statistical Calculation*

For each collocated HIRAS-II—AHI FOV, the brightness temperature difference (BT diff $= \mathrm{BT_{HIRAS}} - \mathrm{BT_{AHI}}$) and the standard deviation of the brightness temperature difference (Std $= \sqrt{\sum_{i=1}^{n}(\mathrm{BT\ diff_i} - \mathrm{BT\ diff_{mean}})^2/(n-1)}$) are counted (n is the number of samples).

A total of 458 pairs of samples are obtained after collocating the HIRAS-II observations with the AHIs from 15 March to 21 April 2022 based on the above matching steps.

*3.5. Matching with MERSI-LL on the Same Platform*

When the HIRAS-II observations are compared with those from the AHI nadir on the geostationary satellite, the matched fields of view are concentrated near the tropical equator, and the dynamic range of the observed brightness temperature is narrow. The HIRAS-II observations will be compared with measurements from one imager that is carried on a polar-orbiting platform to evaluate its observation accuracy on a global scale. However, FY-3E is the first early morning polar-orbiting satellite with a significant observation time difference (even more than 8 h) from the established mid-morning or afternoon orbit satellites. Therefore, the MERSI-LL imager on the same FY-3E platform is chosen to perform the HIRAS-II calibration in this paper, and the MERSI-LL channels 4–7 are spectral matched. Since HIRAS-II and MERSI-LL are on the same polar-orbiting satellite platform and are almost observed simultaneously, the matching process of the two instruments is relatively simple. Only the nadir HIRAS-II FOV is matched to ensure the same observation scene. The spectral matching, spatial matching, and uniformity checking steps are the same as those for AHI. Finally, a total of 12,395 pairs of HIRAS-II and MERSI-LL observation samples were matched over 8 days from 15 to 22 March 2022.

## 4. Results and Discussion

*4.1. Comparison of HIRAS-II with AHI*

There are nine Himawari-8/AHI channels (channels 8 to 16) that are spectral matched with FY-3E/HIRAS-II in the longwave infrared band. The scatter plots of the convolved HIRAS-II observed brightness temperature with AHI measurements in channel 8 to 16 are given sequentially in Figure 4. The horizontal coordinate is the HIRAS-II observed brightness temperature, the vertical axis is the AHI observation, the dashed line is the $y = x$ line, and the solid line is the linear fitting result. Because the matched samples are concentrated near the nadir of the geostationary satellite, the dynamic range of the observed brightness temperature for each channel is narrow, and the value gradually decreases as the peak height of the weighting functions of atmospheric absorption channels increases. The observations of the two instruments are very close, with correlation coefficients higher

than 0.98, and the fitting lines almost coincide with the $y = x$ line in all the channels. The statistical bias and standard deviation for the HIRAS-II—AHI matching samples from 15 March to 21 April 2022 are listed in Table 2. Figure 4 and Table 2 show that all the channels have a slightly positive bias; namely, that the HIRAS-II convolved observations are slightly warmer than AHI with a maximum bias of 0.65 K (channel 9 in the water vapor wing), and the minimum is 0.22 K (window channel 14). The standard deviation of all the channel biases ranges from 0.22 to 0.31 K with small values and little difference between the channels. Water vapor absorption channels 8–10 and ozone absorption channel 12 have relatively larger biases and small standard deviations, while the biases of the window channels (such as channels 14 and 15) are relatively small, and the standard deviations are slightly large. In addition, the closer the peak height of the weighting function is to the surface, the larger the standard deviation is. This is because the value range of the observed brightness temperature of the window channels is relatively larger than that of the absorption channels, so the dispersion is larger in the window channel.

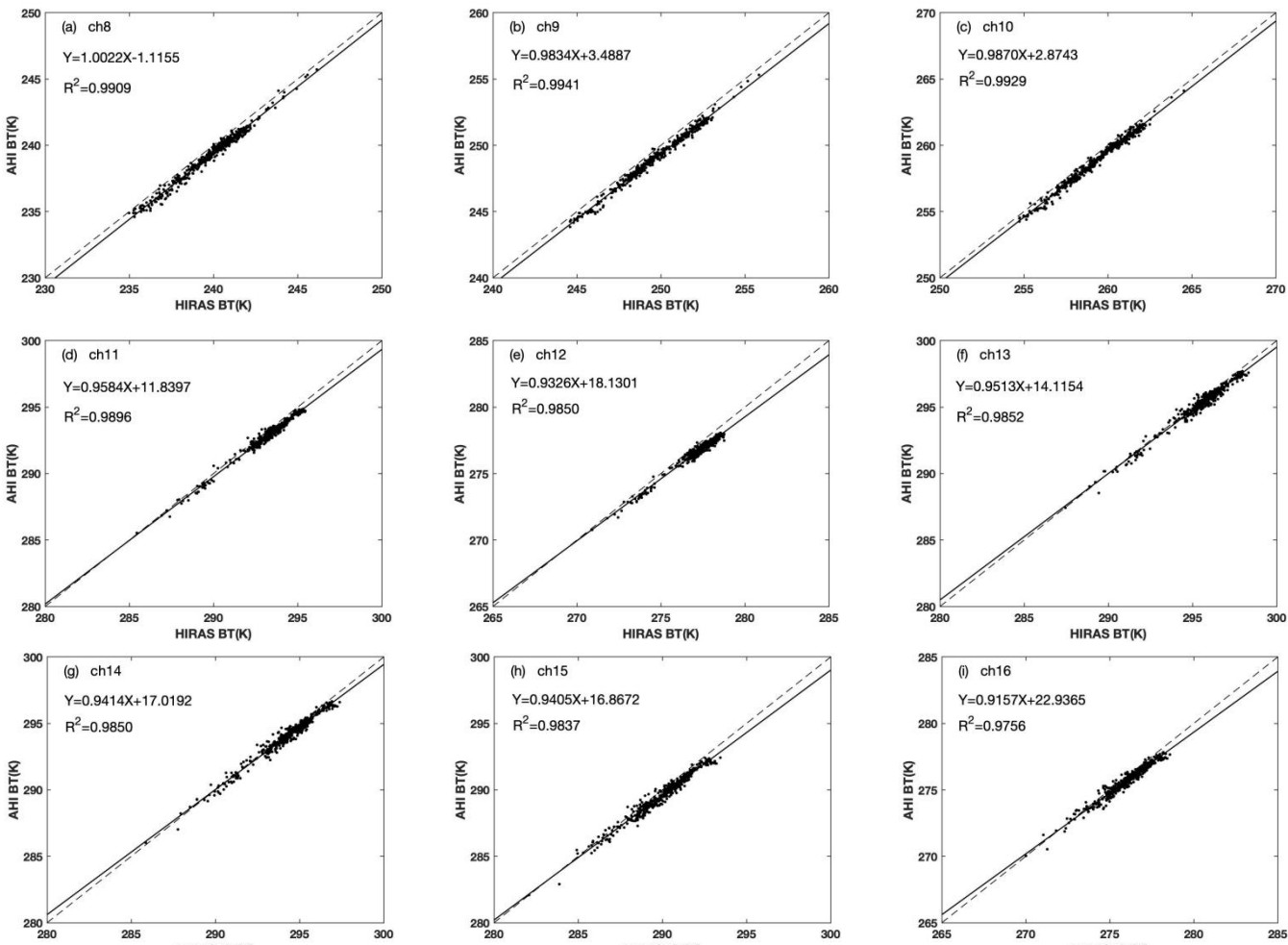

**Figure 4.** Scatterplot of the convolved HIRAS-II brightness temperature and the AHI brightness temperature for channels 8–16. The dashed line is $y = x$, and the solid line is the linear fitting result.

The ideal condition for the realization of the cross-calibration using the simultaneous nadir overpass (SNO) method is that the two instruments observe the same target at the same time through the same atmospheric path. However, the matching condition is appropriately approximated in the actual application to obtain enough samples, and the introduction of the matching threshold may bring uncertainty to the calibration evaluation. For the HIRAS-II and AHI cross-calibration, the observation inconsistency may be partly due to the random errors caused by the differences in observation geometry, scene

uniformity, and observation time. To analyze the possible uncertainties caused by the colocation approximation factors, water vapor channel 8 with the highest peak height of the weighting function is taken as an example. The distribution of the brightness temperature differences of HIRAS-II and AHI with various matching factors: (a) observation geometry, (b) scene uniformity, (c) azimuthal difference, and (d) observation time difference are shown in Figure 5. The larger the value of the horizontal coordinate in Figure 5a denotes the larger the difference in the observation geometric viewpoint. A larger value of the *x*-axis in Figure 5b indicates worse uniformity within the field of view. The larger *x* value of Figure 5c corresponds to the larger difference in the observation azimuth of the two instruments. The solid line shows the linear fit pattern of the brightness temperature differences with the approximation factors. The brightness temperature differences are randomly and uniformly distributed with these factors and do not increase with the decrease in various matching degrees. There is no obvious linear variation characteristic with these matching factors, indicating that the influence of various matching factor differences within their threshold on observation bias can be neglected and that these reasonable matching thresholds bring little uncertainty to the bias assessment. Figure 5e shows the distribution of the brightness temperature differences with the HIRAS-II observed brightness temperature. These is also no significant scene temperature-dependent bias. The results of the remaining channels are similar and omitted. After excluding the random errors caused by the matching factors mentioned above, it can be concluded that the HIRAS-II—AHI brightness temperature differences mainly represent the systematic observation bias of the two instruments.

**Table 2.** Statistics of brightness temperature bias between HIRAS-II and AHI.

|  | AHI | | | | | | | | |
|---|---|---|---|---|---|---|---|---|---|
|  | ch8 | ch9 | ch10 | ch11 | ch12 | ch13 | ch14 | ch15 | ch16 |
| Mean (k) | 0.5780 | 0.6465 | 0.4909 | 0.3726 | 0.5465 | 0.2688 | 0.2274 | 0.3935 | 0.3259 |
| Std (k) | 0.2527 | 0.2398 | 0.2185 | 0.2165 | 0.2171 | 0.2881 | 0.2917 | 0.3061 | 0.2814 |
| Mean HIRAS BT (k) | 240.1 | 250.0 | 259.7 | 293.4 | 277.4 | 295.8 | 294.5 | 290.2 | 276.2 |
| Correlation coefficient | 0.9909 | 0.9941 | 0.9929 | 0.9896 | 0.9850 | 0.9852 | 0.9850 | 0.9837 | 0.9756 |

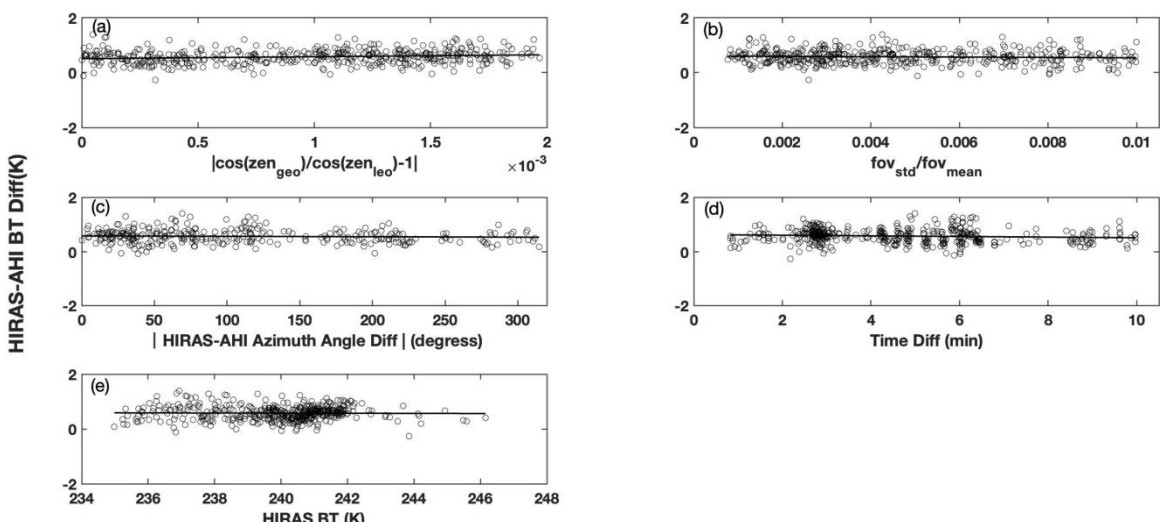

**Figure 5.** Brightness temperature biases between HIRAS-II and AHI of channel 8 varying with (**a**) observation geometry factor, (**b**) scene homogeneity factor, (**c**) azimuth angle factor, (**d**) observation time differences, and (**e**) HIRAS-II observations (the solid line shows the linear fitting result).

*4.2. Comparison of HIRAS-II with MERSI-LL*

The observed biases and standard deviations were counted based on the 12,395 pairs of samples matched by HIRAS-II and MERSI from 15–22 March 2022. The distribution of

the brightness temperature differences (HIRAS-II minus MERSI-LL) with the HIRAS-II observed scene temperature for channel 4–7 are shown in the left subplot of Figure 6 with the vertical coordinates representing the brightness temperature differences and the horizontal coordinates representing the observed scene temperature. The color distinguishes the scene uniformity, and the dashed line gives the mean of the biases. The probability density distributions of the brightness temperature differences are given in the right subplot with the horizontal coordinate as the sample probability density. Channel 4 of MERSI-LL is a water vapor absorption band with a central wavelength of 7.22 μm (peak height of the weighting function 400 hPa), and channel 5–7 are window channels with a central wavelength of 8.55 μm, 10.8 μm, and 12.0 μm, respectively. The dynamic range of the channel 4 target brightness temperature is between 220 and 280 K, and the HIRAS-II measurement is slightly higher than the MERSI-LL observations (the mean bias is 0.6643 K) with a standard deviation of 0.2229 K. The dynamic range of channel 5 is slightly larger at approximately 220–300 K, and the observed brightness temperatures of HIRAS-II and MERSI are close, with a mean bias of 0.0023 K and a standard deviation of 0.3135 K. The dynamic range of channel and 7 target brightness temperature are between 210 and 310 K, and the biases of channel 6 and 7 show a U-shaped change with the increase in the scene temperature, and the biases are smallest (close to 0 K) when the scene temperature is between 250 K and 280 K. Channel 4 has smaller brightness temperature differences at lower scene temperatures (i.e., high latitudes) and relatively larger brightness temperature differences at higher scene temperatures (i.e., low latitudes), especially when the bias increases to 1.2 K near the equator. Although the channel 5 bias takes values of approximately 0 K, the bias dispersion increases as the HIRAS-II observed scene temperature increases. Both channel 6 and 7 have relatively larger brightness temperature differences at lower scene temperatures and higher scene temperatures, and the maximum value is close to 1.75 K. From the right subplot, it can be seen that the probability density distributions of the brightness temperature bias for channels 4–7 all conform to the normal distribution.

It is noteworthy that the brightness temperature differences of the water vapor channel 4 in Figure 6 are obviously positively correlated with the target scene temperature, and the window channels 6 and 7 also have an obvious scene temperature-dependence, while window channel 5 shows no scene temperature-dependent bias. At the same time, AHI water vapor channels 9 and 10—whose spectral positions are close to MERSI-LL channel 4—also do not find bias scene-dependent characteristics. Since HIRAS-II and MERSI-LL are mounted on the same platform, the scene uniformity is the only factor that introduces matching uncertainty into the intercomparison. Figure 7 shows the scatter distribution of MERSI-LL channel 4 (a), channel 5 (b), channel 6 (c), and channel 7 (d) brightness temperature differences (HIRAS-II minus MERSI-LL) with scene uniformity. A larger value of the horizontal coordinate in Figure 7 indicates worse scene uniformity, and the solid line indicates the linear fitting result. The brightness temperature differences of channel 5–7 are uniformly distributed with the scene uniformity and do not have linear variation characteristics (in Figure 7b). However, the brightness temperature differences of channel 4 show an obvious linear trend with the scene uniformity, and the biases gradually decrease as the scene uniformity worsens (in Figure 7a). Combined with the scatter color of channel 4 in Figure 6, the scene uniformity is relatively poor (yellow) in the high latitudes with a low brightness temperature, and the scene uniformity is good in the low latitudes with a high brightness temperature. This is because the underlying surface in the field of view varies greatly in the polar region when the instrument is scanning with the same spatial resolution and swath, especially the Arctic has greater underlying surface variability due to the presence of different surface types (e.g., land, snow, ocean, glacier, etc.) with higher variability in absolute temperature. Theoretically, the bias is smaller when scene uniformity is better. However, Figure 6 shows that the scene uniformity gradually improves with the increasing scene temperature, while the bias increases instead. This indicates that the scene uniformity is not the cause of the scene temperature-dependent bias.

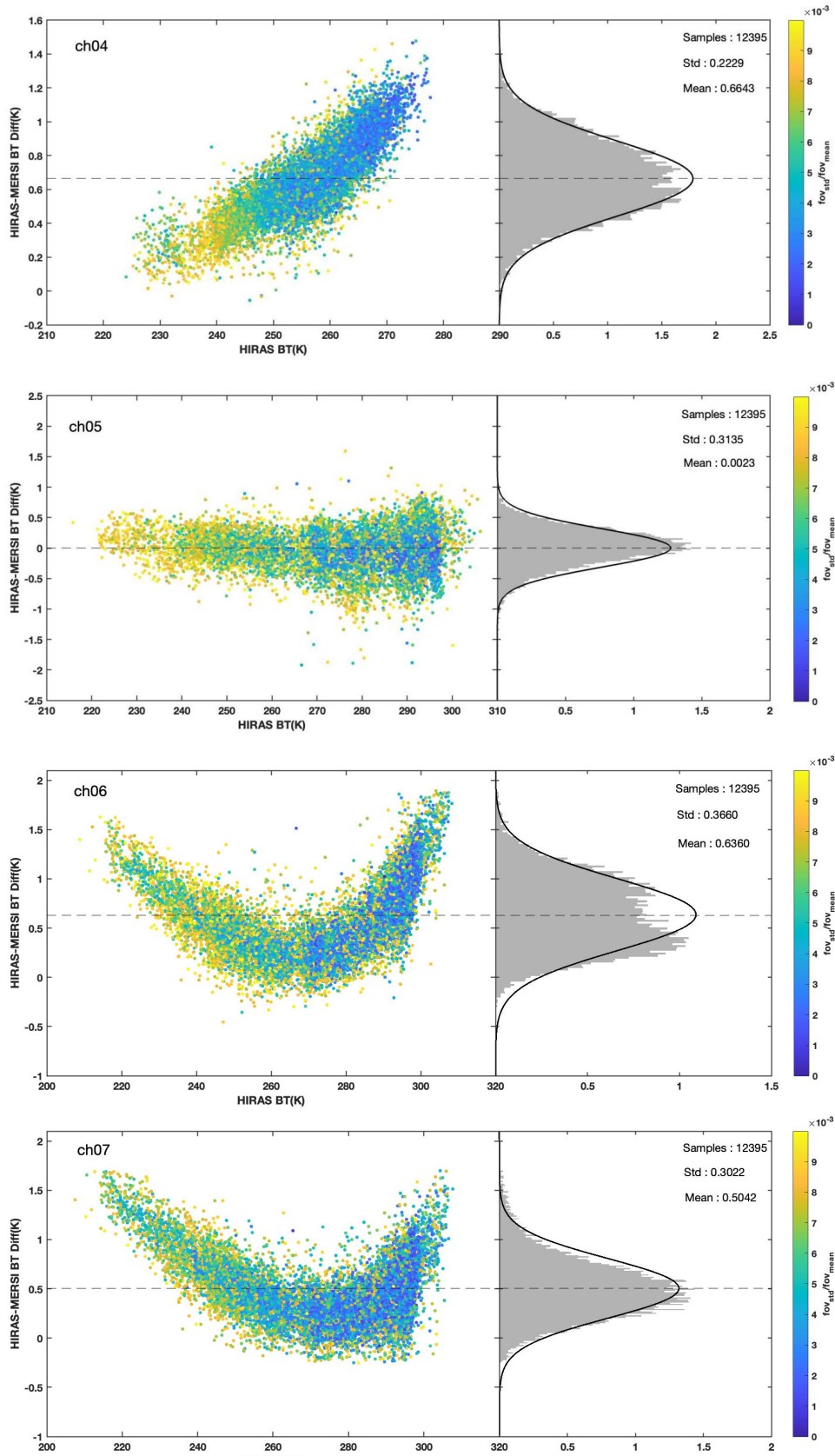

**Figure 6.** (left) Scatterplot of the HIRAS-II–MERSI-LL BT bias versus the scene temperature and (right) the histogram of the BT differences. The dashed line indicates the mean value. The color indicates the scene homogeneity.

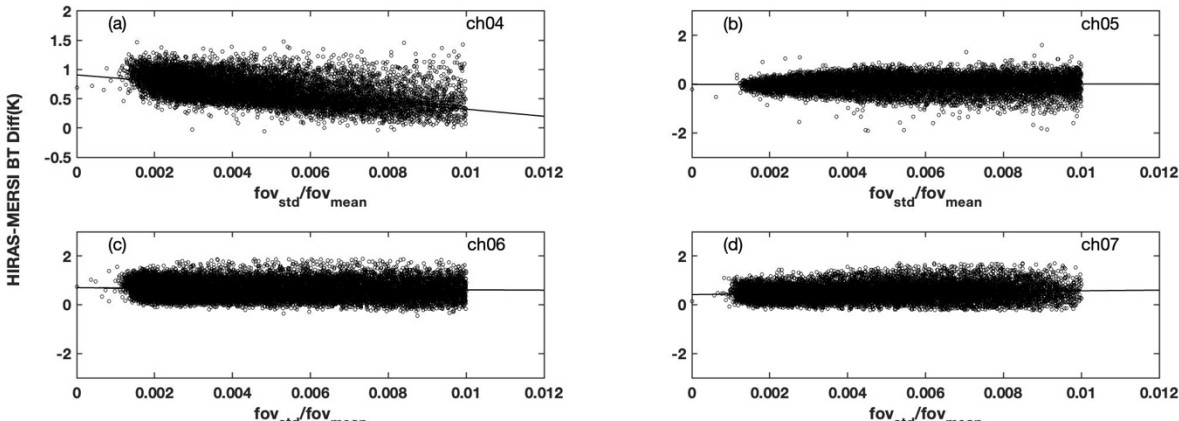

**Figure 7.** Scatterplot of brightness temperature biases between HIRAS-II and MERSI-LL varying with scene homogeneity of (**a**) channel 4, (**b**) channel 5, (**c**) channel 6, and (**d**) channel 7 (the solid line shows the linear fitting result).

The main factors that may cause spaceborne radiation imager calibration errors mainly include blackbody emissivity and spectral response function instrument nonlinearity. MERSI-LL channel 4 is located in the wing area of the water vapor absorption band, and a very small drift in the spectral response function can also lead to a temperature-dependent bias in the scene. However, to date, there have been no specific references about the design of the black bodies, the calibration system, and so on of these two instruments onboard FY-3E. These matters require further study in the future.

The day-to-day variations in the mean biases and standard deviations are counted based on the HIRAS-II–MERSI-LL matched samples. The daily mean biases of channel 4 range from 0.64 to 0.68 K, and the standard deviations are all approximately 0.2 K. The daily mean biases of channel 5 range from −0.02 to 0.02 K, and the standard deviations are approximately 0.3 K. The daily mean biases of channel 6 range from 0.62 K to 0.64 K, and the standard deviations are approximately 0.35 K. The daily mean biases of channel 7 range from 0.48 K to 0.51 K, and the standard deviations are approximately 0.3 K. The biases of the two instruments vary minimally and almost constantly over a period of time, indicating that the performance of the HIRAS-II instrument is stable.

## 5. Conclusions

To assess HIRAS-II's on-orbit observation quality, the geometrically, temporally, and spatially matched scene homogeneous HIRAS-II hyperspectral observations were convolved to the longwave infrared channels corresponding to the Himawari-8/AHI and FY-3E/MERSI-LL from 15 March to 21 April 2022, and their brightness temperature deviation characteristics were statistically calculated in this paper. The matching samples of HIRAS-II and AHI are concentrated near the equator, and the spectral matching channels are longwave infrared channel 8 to channel 16 onboard the same polar orbiting satellite platform FY-3E. The matching samples of HIRAS-II and MERSI-LL are evenly distributed all over the world with spectral matching channels 4 to channel 7. The following conclusions can be made based on this analysis:

1.　The HIRAS-II on-orbit observed brightness temperatures are slightly warmer than the AHI observations, with a small positive bias in all the matched channels. The brightness temperature scatters of both observations are distributed near the $y = x$ line with a correlation coefficient higher than 0.98 in all channels. The biases of water vapor channels 8–10 and ozone absorption channel 12 are relatively large, with a maximum of 0.65 K (channel 9 in the water vapor wing), and the biases of the window channels are relatively small, with a minimum of 0.22 K (channel 14). The standard deviations for all channels are small (0.22–0.31 K) and there is little difference between the channels.

2.  The thresholds chosen for the colocation approximation factors (e.g., observation geometry angle, field of view uniformity, observation azimuth, and observation time) when matching the HIRAS-II with AHI contribute little and negligible uncertainty to the bias assessment, so the difference between the two observed radiations is considered to be mainly from the systematic bias of the two instrument measurements.

3.  Since HIRAS-II and MERSI-LL are mounted on the same platform, the scene uniformity is the only factor introducing matching uncertainty in the intercomparison. The mean brightness temperature bias (HIRAS-II minus MERSI-LL) of the MERSI-LL water vapor channel 4 is 0.66 K with a standard deviation of 0.22 K. To window channel 5, the observations of both instruments are very close, with a mean bias of 0.002 K and a standard deviation of 0.31 K. Both channel 6 and 7 have relatively larger brightness temperature differences at lower scene temperatures and higher scene temperatures, with a mean bias of 0.63 K (the standard deviation is 0.36 K) and 0.5 K (the standard deviation is 0.3 K), respectively.

4.  The biases of MERSI-LL channel 4 are obviously positively correlated with the target scene temperature. The biases of channel 6 and 7 show a U-shaped change with the increase in the scene temperature, and the biases are smallest (close to 0 K) when the scene temperature is between 250 K and 280 K. The statistical characteristics of the HIRAS-II–MERSI-LL difference vary minimally and almost constantly over a period of time, indicating that the performance of the HIRAS-II instrument is stable.

As a final note, we just found the phenomenon of bias distribution, which is not yet fully explained due to lack of relevant references. Therefore, we will use NWP data, double-difference method to further evaluate the accuracy of HIRAS-II in future studies.

**Author Contributions:** Conceptualization, L.G. and H.C.; methodology, L.G.; software, H.C.; validation, H.C.; formal analysis, L.G.; investigation, H.C.; resources, L.G.; data curation, H.C.; writing—original draft preparation, H.C.; writing—review and editing, H.C.; visualization, H.C.; supervision, L.G.; project administration, L.G.; funding acquisition, L.G. All authors have read and agreed to the published version of the manuscript.

**Funding:** This work was supported by the National Natural Science Foundation of China under grant no. 41975028.

**Data Availability Statement:** The HIRAS-II and MERSI-LL Level 1 data can be obtained at the Chinese Feng Yun satellite remote sensing data service network (http://data.nsmc.org.cn accessed on 11 April 2022). Himawari-8/AHI radiation data obtained from the Japan Earth Observation Data Center (https://www.eorc.jaxa.jp/ptree/index.html accessed on 11 April 2022).

**Acknowledgments:** The authors would like to thank the editor and reviewers for their helpful comments on the manuscript.

**Conflicts of Interest:** The authors declare no conflict of interest.

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
