# Peer review of "Assessing FY-3E HIRAS-II Radiance Accuracy Using AHI and MERSI-LL"

_remotesensing, doi:10.3390/rs14174309_

Round 1

Reviewer 1 Report

1. In order to maintain the consistency of the specific entries' naming, the authors had better use the HIRAS-II and MERSI-LL instead of HIRAS and MERSI, respectively. 

2. The 4th item in conclusion part, the expression "may be" is not rigorous.

3. It is preferable not to show Fig. 6 across pages.

4.The authors should pay attention to the paper typesetting, for example, there is too much blank in front of and behind the Fig. 7, which should be removed.

5. In addition to HIRAS-II and AHI, the article also analyzes a lot of MERSI-LL which is not reflected in the title, and the paper’s title should take it into consideration.

6. This paper is logical and innovative, but the expression should be improved.

Author Response

Thank you for taking time out of your busy schedule to review the manuscript. Please see the attached reply report.

Reviewer 2 Report

Please see the attached review report.

Author Response

(The authors gave the same response as above.)

Reviewer 3 Report

The focus of this manuscript is the radiometric performance assessment of HIRAS-â…¡ on-board FY-3E using AHI onboard Himawari-8 as reference. It is useful and easy to read. Data and methods are clearly described, and the results and analysis are reasonable. 

Following is my comments:

1 What is the “MTSAT” in line 85. When abbreviations appear in the text for the first time, please indicate the full name.

2 What’s the “microlight” in line 104 and 105?

3 I suggest that the evaluation method introduced in line 203 – 207 should be a separate sub-section. It is not only used for cross-comparison of HIRAS-II and AHI.

4 The analysis on the large brightness temperature differences of the water vapor channel 4 is confuse for me. Why does a non-ideal blackbody cause this large deviation, When the emissivity of blackbody is known or calibrated before launch.

Author Response

(The authors gave the same response as above.)

Reviewer 4 Report

Dear authors,

Congratulations on a well-written and concise manuscript "Assessing FY-3E HIRAS-II Radiance Accuracy Using AHI 2 onboard Himawari-8". I haven't found any major scientific/methodological concerns, I only have some minor questions and remarks that are listed below.

* L. 153 - 155: For the casual reader, it would be good if this sentence is also added to the Figure 2 caption.

* L. 166: Depending on the surface and season, it could be that a time difference of < 10 minutes still gives an additional difference of several tenths of a degree. Especially in mid-morning during spring/summer, the air temperature can already increase by 4-5 K within 1 h, so for surface BT this will be even more. Can you give any thoughts on this?

* L. 182 - 183: I assume that with the homogeneity criterion the issue of fractional pixel coverage of AHI over the HIRAS-II FOV is covered? Please indicate this in the text.

* L. 275: Figure 5 (c) --> Figure 5 (c)

Author Response

(The authors gave the same response as above.)
